SciPost Physics

Submission

# Search for upward-going air showers with the fluorescence detector of the Pierre Auger Observatory

Emanuele De Vito[1,2] for the Pierre Auger Collaboration[3]⋆

**1** Università del Salento, Dipartimento di Matematica e Fisica "E. De Giorgi", Lecce, Italy
**2** INFN, Sezione di Lecce, Lecce, Italy
**3** Observatorio Pierre Auger, Av. San Martín Norte 304, 5613 Malargüe, Argentina
(Full author list: https://www.auger.org/archive/authors_2021_09.html)
*email: spokespersons@auger.org

May 11, 2022

*16th International Workshop on Tau Lepton Physics (TAU2021),*
*September 27 – October 1, 2021*

## Abstract

The fluorescence detector (FD) of the Pierre Auger Observatory is sensitive to upward-going air showers for energies above $10^{17}$ eV. Given its operation time and wide field of view, the FD has the potential to support or constrain the recent "anomalous" observations by the ANITA detector, interpreted as upward-going air showers of unexplained nature.
We have used 14 years of data collected by the FD to search for upward-going showers using a set of quality selection criteria defined using 10% of the full data sample. To distinguish candidates from false positives, calculate the exposure and obtain the expected background, dedicated simulations for signal (upward-going events) and background (downward-going events) have been performed. Results of the analysis after unblinding the data set are presented.
Finally, the exposure and sensitivity for the specific scenario of a signal being ascribed to tau lepton decay are calculated and the corresponding upper limits are shown as a function of primary energy and in different zenith angle ranges.

# 1   Introduction

The Pierre Auger Observatory is designed to detect extensive air showers produced by Ultra-High Energy Cosmic Rays (UHECRs). It is located near the city of Malargüe, Argentina, and is composed of a surface detector (SD) array of 1660 water-Cherenkov stations and 27 telescopes grouped in four sites forming the fluorescence detector (FD). The SD stations sample the density of the secondary particles of the air shower at the ground while the FD observes the longitudinal development of the air shower detecting the fluorescence and Cherenkov light emitted from the interaction of secondary particles with the atmosphere [1].

The Pierre Auger Observatory is the largest and most precise detector of UHECRs. In addition, it has the possibility to detect UHE neutrinos with the SD looking at inclined events [2] (zenith angle $\theta$ between 60° and 90°) or slightly upward-going events (90° < $\theta$ < 95°) in the Earth-skimming channel [3]. The FD can also in principle detect upward-going air showers covering a wider zenith range. Such a search would certainly extend our understanding of the Standard Model even testing some exotic Beyond the Standard Model (BSM) scenarios [4]. Moreover, the observation by ANITA of two anomalous upward-going events not consistent with reflected pulses from cosmic ray showers [5], encouraged many collaborations to search for possible anomalies in the expected neutrino fluxes.

Given its operation time and wide field of view, the FD has the potential to support or constrain these recent "anomalous" observations by the ANITA detector. The two events were detected during the first and the third flights of ANITA with an elevation angle of 27.4 ± 0.3° and 35.0 ± 0.3°, respectively, and energies above ∼ 0.2 EeV. The energies and the elevation angles of these two events appear challenging to reconcile with the predictions of the standard model of particle physics, so a confirmation or a constraint from a different experiment would be of particular interest.

We have used 14 years of data collected by the FD to search for upward-going showers using a set of quality selection criteria defined using 10% of the full data sample. Simulations have been used to study the FD trigger efficiency to an upward-going signal and our capability to distinguish candidates from false positives. Tau leptons have also been simulated to test the specific scenario of a tau-initiated air shower. Simulations will be described in detail in section 2. In section 3, the analysis will be presented with a focus on the final cut used to discriminate between upward and downward-going events both in data and simulations. In section 4, the details about the exposure calculation will be given for the generic search of an upward-going signal and for the tau-specific scenario. Finally, in section 5, the final result after the unblinding and the corresponding upper limits will be presented.

# 2   Simulations

An upward-going air shower can be observed if a particle emerging from the Earth interacts or decays in the atmosphere or in the rock right below the Earth crust. To estimate the FD capability of triggering such an event, simulations are of crucial importance.

Upward-going protons have been simulated with $\log_{10}(E/\text{eV}) \in [16.5, 18.5]$ and a zenith angle $\theta \in [110°, 180°]$. The first interaction point $H_{\text{fi}}$ has been fixed in the range $[0,9]$ km above the ground altitude of the Observatory (∼ 1400 m a.s.l.) as showers that start at higher

altitudes are naturally less likely to be triggered because they tend to be further away. Protons have been chosen because they can be easily adapted to fit any interesting scenarios such as neutrinos or BSM particles [6].

Downward-going events are a background source for this search. Indeed if the core of a downward-going event is located behind the FD telescope (figure 1 left), the generated shower gives an upward-going track in the camera mimicking an upward shower in the center of the array. To calculate the expected background for this analysis, events with $\log_{10}(E/\text{eV}) \in [17, 20]$ and a zenith angle $\theta \in [0°, 90°]$ have also been simulated.

Finally, to evaluate the capability of the FD to trigger a tau-initiated shower, tau leptons have been simulated starting inside the Earth within 50 km before the exit point or in the atmosphere below an altitude of 9 km, with $\log_{10}(E/\text{eV}) \in [16.5, 20]$. Figure 1 (right) shows all possible outcomes for the generated tau. Cases 3 and 5 in which tau-lepton decay occurs in the FD field of view have been used to calculate the expected trigger rate for a tau air shower. Tau leptons have been simulated with TAUOLA [7] considering all the decay branches with only $e^{\pm}$, $\pi^{\pm}$, $\pi^0$, $K^{\pm}$ and $K^0$ meaningfully contributing to the energy of the resulting atmospheric air shower.

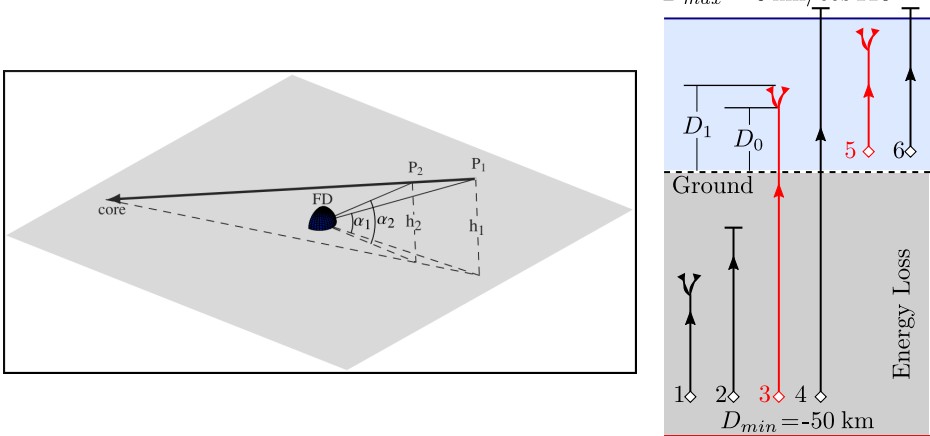

Figure 1: (Left) Graphical representation of a downgoing event landing behind the FD site that can mimic an upward-going event. (Right) Representation of tau simulations. Tau decays which may trigger the FD are indicated in red.

## 3  Analysis

A blind analysis has been performed using 10% of FD data to study the background due to mis-reconstructed downward-going events [8]. Moreover, upward-going laser shots, used by the Collaboration for atmospheric monitoring, represent another possible source of background. They are mainly shot by two facilities located in the middle of the array and by four LIDARs located at each FD site. Lasers are mostly fired vertically, and the large majority are rejected based on their known timestamp. However, it may happen that some laser shots leak into the data sample as genuinely upward-going events. An algorithm has been developed to identify and reject the remaining laser shots by exploiting the time of each event and its position inside the array.

A profile-constrained geometry fit has been applied to the remaining sample, testing if any possible upward-going geometry can explain the event. In that case, downward geometries

have been tested too. A variable $X = \arctan(-2\log(L_{\text{down}}/L_{\text{all}})/50) \cdot 2/\pi$ has been defined to compare the two reconstructions where $L_{down}$ is the likelihood of the downward reconstruction, while $L_{all}$ is the maximum likelihood between the upward and the downward reconstruction. This variable has been defined so that an event with $X = 0$ is more likely a downward-going event, while if $X = 1$ the upward reconstruction is favoured.

Figure 2 shows the distribution of the $X$ variable for the events in the signal and background simulations as well as for the 10% data sample. According to this distribution, a cut value of $X = 0.55$ has been set with a survival rate for the signal of about 60% and an expected background of $\sim 0.5$ events in the full data sample.

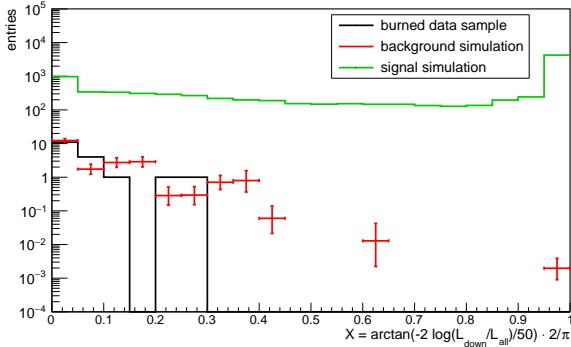

Figure 2: Distribution of the $X$ variable for the events from 10% of data (black), background simulations (red), signal simulations (green). We have set a cut value on $X = 0.55$ to discriminate between background and signal region with a survival rate for the signal of about 60% and an expected background of $\sim 0.5$ events in the full data sample.

# 4 Exposure

When simulating an upward-going air shower, the height of the first interaction point $H_{\text{fi}}$ can significantly change the trigger efficiency of the FD. For this reason a double differential exposure has been calculated as

$$\frac{d\varepsilon}{dH_{\text{fi}}}(E_{\text{cal}}, H_{\text{fi}}) \simeq 2\pi \cdot S_{\text{gen}} \cdot \Delta T \cdot \sum_{i} \eta(E_{\text{cal}}, \cos\theta_i, H_{\text{fi}}) \cdot \frac{1}{\Delta H_{\text{fi}}} \cdot \cos\theta_i \cdot \Delta\cos\theta_i \qquad (1)$$

where $E_{\text{cal}}$ is the energy released by the shower in the air, $S_{\text{gen}}$ is the surface area of generation (a square of $100 \times 100$ km$^2$), $\Delta T$ is the 14 years of operation of the FD, $\eta$ is the fraction of events passing the selection and $\theta$ is the zenith angle. For this study we limited our analysis to events with $\theta > 110°$ motivated by the elevation of the two "anoumalous" events observed by ANITA.

Figure 3 (left) shows this exposure based on upward-going proton simulations as a function of $E_{\text{cal}}$ and $H_{\text{fi}}$. As expected the FD exposure increases with energy being very suppressed for energies below $10^{17}$ eV.

Tau simulations have also been used to calculate the number of tau leptons decaying in the field of view of the FD. By folding this with $d\varepsilon/dH_{\text{fi}}$, the double differential exposure to tau lepton air shower is derived. Finally for the scenario of tau lepton production, the FD exposure is calculated as a function of the initial energy of the tau by integrating over $E_{\text{cal}}$ and $H_{\text{fi}}$. The right panel in figure 3 shows this exposure for the whole zenith range and for three different

intervals. As expected, the exposure increases with energy and is higher for inclined events (lower zenith angles). For energies above $10^{18.5}$ eV since no simulations are available, the value obtained at $10^{18.5}$ eV is used as a lower bound of the exposure. Further work to extend the energy range is planned.

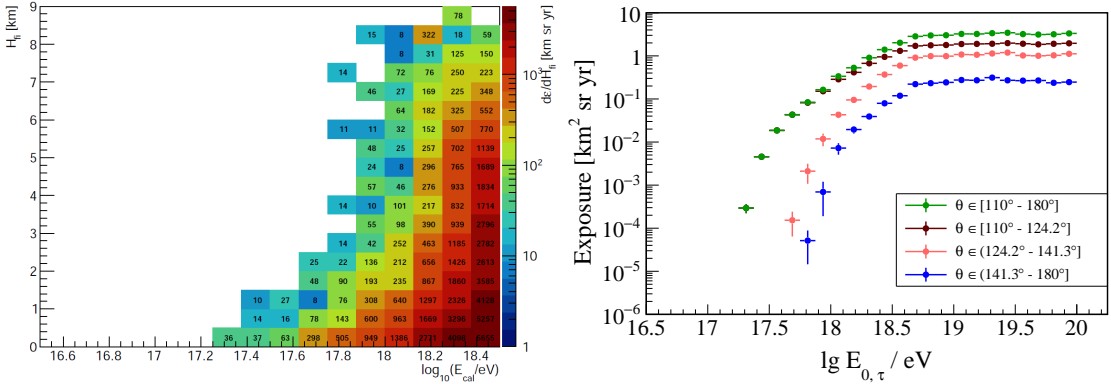

Figure 3: (Left) Double differential exposure with $\log_{10}(E_{cal}/eV)$ on the x-axis and the height of first interaction on the y-axis for upward-going events. (Right) FD exposure for upward-going showers induced by tau-decay as a function of the lepton energy ($E_{0,\tau}$), for the whole zenith range (green) and for three different zenith intervals.

# 5  Unblinding and upper limits

The unblinding procedure led us to only one event passing all the quality cuts including the cut on $X$, the discriminating variable. This result is compatible with the expected background and, as a consequence, an integral upper limit to the flux of upward-going air showers (95% confidence level) can be set at $3.6 \cdot 10^{-20}$ cm$^{-2}$sr$^{-1}$s$^{-1}$ $\left(8.5 \cdot 10^{-20}$ cm$^{-2}$sr$^{-1}$s$^{-1}\right)$ by weighting the exposure with $E_{cal}^{-1}$ $\left(E_{cal}^{-2}\right)$.

Based on this result, also the corresponding upper limit for the flux of upward-going tau leptons has been set for two different injection spectral indices. Since the FD exposure is higher for lower zenith angles, the tau limit has been set for three zenith ranges between 110° and 180°. The most horizontal zenith angles provide the best limits. Moreover, the weighting of the exposure with $E_{0,\tau}^{-1}$ (figure 4 left) led to a lower limit for high energies with respect to the one weighted with $E_{0,\tau}^{-2}$ (figure 4 right). This is related to the shape of the exposure which is assumed to be flat above $10^{18.5}$ eV.

# 6  Conclusion

A search for upward-going air showers with the fluorescence detector of the Pierre Auger Observatory has been performed. Only one event, consistent with the expected background, has been observed, and an upper limit to the flux of upward-going air showers has been set. The hypothesis of a shower initiated by a tau lepton has also been investigated, and a differential upper limit as a function of the tau energy for two injection spectra and three different zenith intervals has been calculated.

In absence of published exposure values related to the two ANITA events, we cannot directly

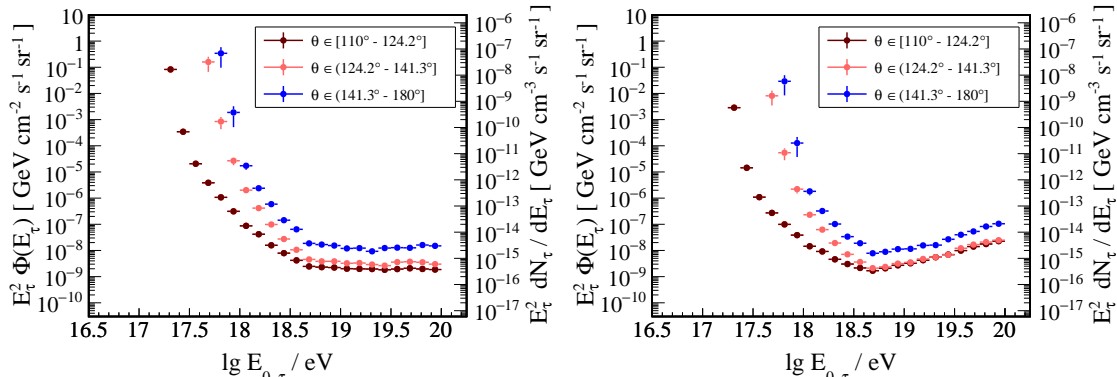

Figure 4: Upper limit set with the hypothesis of 1 background event and a spectral index equal to −1 (left) and −2 (right) for three different zenith intervals.

compare our results to that of ANITA. This result is still of interest to test any possible BSM scenario of a particle decaying or interacting above the Earth's crust and generating a tau lepton.

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
