# Peer review of "Search for upward-going air showers with the fluorescence detector of the Pierre Auger Observatory"

_SciPost Physics Proceedings_

## Round 1 · Referee Report · Michele Doro (Referee 1) · 2022-5-3

Report

Dear authors

apologies for the delay in the review. The proceedings are well written, clear and the results and methods are sound. I only have few minor remarks and curiosities: * you did not mention how did you choose the 10% test dataset, and whether the acceptance was constant during the 14y operation * the justification of X=0.55 is rather vague. Can't you estimate a false negative/false positve/power of your test? * when mentioning the figures you should mention left/right

That's it

Requested changes

  • when mentioning the figures you should mention left/right

  • validity: ok
  • significance: ok
  • originality: good
  • clarity: high
  • formatting: good
  • grammar: excellent

Author:  Emanuele De Vito  on 2022-05-11  [id 2455]

(in reply to Report 1 by Michele Doro on 2022-05-03)

Dear reviewer,
thanks for your comments and suggestions.
The 10% data set has been chosen by requiring the presence of the number 1 in the millisecond GPS timestamp. Therefore the events have been almost "randomly" chosen from the entire data sample. The acceptance of our detector was not constant during the 14 years of operation but we have used Monte Carlo simulations that reproduce the realistic DAQ condition of the detector with high accuracy (in bins of 10 minutes). This will guarantee that simulations and data are consistent at large extent.
The value X=0.55 has been set making a scan through the whole X range and selecting the value that maximize our exposure (i.e. minimize our upper limit). Our upper limit was set with a confidence level of 95% (Feldman and Cousins), we have added a line in the text to specify it.
Finally we also modified the inline reference to figures by adding a left/right as you required.
Best regards.
Emanuele De Vito

---

## Editorial Decision

editorial_decision: